# Impact of Superiors’ Ethical Leadership on Subordinates’ Unethical Pro-Organizational Behavior: Mediating Effects of Followership

**DOI:** 10.3390/bs13060454

**Published:** 2023-05-31

**Authors:** Chulwoo Kim, Chulwoo Lee, Geon Lee

**Affiliations:** 1Department of Public Administration, Gachon University, Seongnam 13120, Republic of Korea; cwkim@gachon.ac.kr (C.K.); chester@gachon.ac.kr (C.L.); 2Department of Public Administration, Hanyang University, Seoul 04763, Republic of Korea

**Keywords:** ethical leadership, followership, unethical pro-organizational behavior, ethics, social learning theory, public officials, government

## Abstract

This study examined the impact of superiors’ ethical leadership on subordinates’ unethical pro-organizational behavior (UPB) and the mediating effects of followership. The research subjects were officials from the ten central departments of the South Korean government, and a cross-sectional survey was conducted among them. Overall, 404 questionnaires were used in the empirical analysis. Multiple regression analysis and Hayes Process Macro were used to validate the research hypotheses, which examined the relationship among ethical leadership, followership, and UPB. The results are as follows: First, the relationship between ethical leadership and followership was statistically significant. Second, the study showed that followership had a statistically significant effect on UPB but not ethical leadership. Third, testing the hypotheses regarding the mediating effect of followership on the relationship between ethical leadership and UPB revealed statistically significant results. This study confirms that followership significantly influences UPB and suggests that ethical leadership is an important precedent factor of followership. The study concludes with the theoretical and practical implications of these findings, along with the study’s limitations.

## 1. Introduction

Ethics has been a longstanding concern in organizational studies, but it is only in recent years that scholars have focused on ethical issues from a leadership perspective [1,2]. A growing body of research has shown that ethical leadership can effectively address subordinates’ unethical behavior [3,4,5,6]. However, the relationship between ethical leadership and unethical yet pro-organizational behavior requires further examination. While unethical behavior is typically at odds with an organization’s interests, there are instances where subordinates may engage in unethical actions that they believe benefit the organization (i.e., unethical pro-organizational behavior or UPB) [7].

This issue is particularly relevant to public organizations. A firmly group-centered culture often emphasizes individuals’ willingness to sacrifice for the organization and view its success or failure as their own [8,9]. Consequently, public officials may engage in unethical behavior with pro-organizational intentions, such as concern for the organization’s reputation or a sense of political responsibility to the organization’s leader. Therefore, it is essential to determine whether ethical leadership has the same effect on UPB in public institutions as it does in other settings.

Nonetheless, the leadership influence process can only be completed with an understanding of followership. For followership, the traits and behaviors of individuals in relation to a leader(s) and their reactions to attempted leadership are also a vital part of the leadership process [10,11]. Followers’ observational learning determines the role model of an ethical leader [12]. The acceptance of ethical leadership varies depending on the type of followership among subordinates. Operating an ethical organization requires the combined efforts of leaders and followers, and ethical behavior (ethical behavior in an organization refers to actions and decisions made by individuals within the organization aligned with moral and societal standards [13]; it involves making choices that consider the well-being of all stakeholders, including employees, customers, shareholders, and the public [14]; ethical behavior also requires complying with legal and regulatory requirements and responsibility in all aspects of work) in organizations should be aligned with ethical leadership and followership [4,15,16]. Therefore, examining how subordinates’ followership characteristics impact UPB in connection with ethical leadership is both theoretically and practically significant.

We address the following research questions: What is the relationship between superiors’ ethical leadership and subordinates’ UPB, and how does subordinates’ followership mediate this relationship? Our model (Figure 1) shows that ethical leadership raises followership, promoting subordinates’ UPB.

Brown et al. (2005) [15] used social learning theory as a theoretical foundation for constructing an ethical leadership theory. According to social learning theory, individuals learn to model and imitate others’ behavior with social modeling and observational learning [17,18]. Ethical leadership by superiors can provide a positive social model for their subordinates. Subordinates can observe and learn from superiors demonstrating ethical values and behaviors. According to the same theory, superiors’ ethical leadership is expected to suppress subordinates’ UPB by emphasizing ethical behavior and presenting exemplary behavior.

According to Kelley’s followership theory, followership consists in independent critical thinking and active engagement [19]. When independent critical thinking responds to the ethical leadership of a superior, individuals judge and act based on moral principles and values that may appear high. In addition, these individuals seek autonomous and moral behavior and adhere to organizational interests and societal standards. At the same time, people with high followership do not just make judgments but actively engage based on those judgments. Therefore, individuals high in followership are expected to be less likely to engage in UPB.

The findings of this study contribute to the literature on leadership and organizational ethics in multiple ways. First, this study contributes to existing research on ethical leadership by introducing the concept of followership and examining its role in promoting ethical consciousness. Identifying the mutual influence between leaders and followers, especially the role of followers in promoting ethical conduct within an organization, is a crucial gap in this field [20,21].

Second, this study was among the first to investigate the mediating role of followership in leadership and UPB. Contrary to the traditional leader-centric approach in the leadership literature, this study takes a role-based followership approach and examines the development of organizational ethics by considering subordinates and their characteristics.

Third, this empirical study was conducted in a public organizational setting. Existing research on ethical consciousness has mostly focused on the corporate sector, and a detailed empirical analysis of ethical consciousness in the public sector is needed [22,23,24]. It is essential to acknowledge the differences in characteristics between public and private organizations and to examine them separately. With a critical analysis of ethical leadership and unethical behavior, this study aims to expand the scope of research beyond corporate-oriented research and provide valuable insights for promoting ethical behavior in the public sector.

This study presents the theoretical background of the main variables used in this research in the following section. It then proposes a research model and hypotheses based on a literature review. Multiple statistical methods are used to test the hypotheses. Finally, the results are presented; theoretical and practical implications are discussed; and limitations and future research directions are suggested.

## 2. Theoretical Background

### 2.1. Unethical Pro-Organizational Behavior

To explain unethical behavior, Umphress et al., defined UPB as “behavior that intentionally promotes the effective activities and functions of an organization and its members while violating sorely recognized standards related to appropriate social values, customs, and laws [7] (p. 771).” The “Boundary Conditions” for UPBs include clarity of intention, not considering the consequences of actions, and prioritizing organizational interest. Therefore, behaviors that do not satisfy these three conditions are not defined as UPBs [25] (p. 622).

The concept of UPB consists of unethical behavior and pro-organizational behavior. Unethical behavior denotes illegal or socially unacceptable behavior, including criminal behavior (e.g., attempts to make an organization look good by telling a lie, exaggerating the positive attributes of products and services, and concealing or omitting negative information related to products and services). Pro-organizational behavior refers to the overall behavior that helps organizations, regardless of whether such behavior is officially part of the job description or an order from a superior [25].

Differences between similar concepts must be identified to understand UPB clearly. Ilie (2012) argued that UPB should be distinguished from organizational deviance, positive deviance, pro-social rule breaking, and illegal corporate behavior [26]. UPB has different reference points for each concept and violation process. On the one hand, it violates social norms, and on the other, unlawful or illegal corporate behavior disregards the law’s effects on the organization’s benefits. In addition, the difference is identified by showing that organizational deviation is a violation of organizational norms that harm the organization and that positive deviation does not follow organizational norms for the organization’s own benefit.

Mishra et al., differentiated among concepts similar to those of UPB. The scope of benefits goes beyond the organization, is obligatory, and includes ethical behavior compared with UPB [27]. In addition, UPB can be characterized as a sub-concept when comparing organizational misbehavior and unethical decision making [28,29]. Compared with pro-self unethical behavior, UPB can distinguish whether it is individuals or organizations that want to receive benefits. Therefore, it clearly represents a unique phenomenon that existing concepts cannot fully explain. Accordingly, UPB can be defined as out-of-duty behavior that violates social norms for the benefit of an organization and its members.

### 2.2. Ethical Leadership

Ethical leadership expresses individual behavior and behavior in relationships with others by appropriately standardizing them while inducing corresponding behavior from followers with mutual decision-making processes, reinforcement, and discussion [13]. First, the standardization of appropriate behavior refers to a dynamic aspect: the contents of ethics (such as everyday norms) and its elements change according to the time and place. Furthermore, this notion implies that the standards of behavior change according to society or a specific organization [30]. The decision-making process is important because ethical leaders must make ethical decisions that produce good results and publicly assume responsibility for their choices by adhering to the principle of fairness [31]. Moreover, strengthening pertains to setting certain ethical standards within the capacity of ethical leaders and providing rewards and punishments for ethical and unethical behavior [23]. A mutual dialogue focuses on ethical issues to promote the interests of subordinates; additionally, facilitating discussion among members instills fairness in procedures or interpersonal relationships [31]. This study follows the definition by Brown and Treviño, based on which ethical leadership is “demonstrating a model of how leaders themselves ethically behave while promoting two-way communication and managing decision making to encourage followers to act ethically” [32]. Ethical leaders form guidelines or organizational norms that subordinates follow, making organizations more trustworthy. In a high-trust environment, subordinates develop a sense of attachment to their organization [33].

Meanwhile, Treviño et al., one of the leading scholars on ethical leadership, emphasized that leaders must first be equipped with the characteristics of a moral administrator and moral person to be recognized as ethical by members [34]. Therefore, ethical leaders must demonstrate their behavior as moral persons and administrators. Specifically, sincere ethical leaders must possess the elements of a moral person as well as a moral manager. Brown et al., summarized the concept of ethical leadership in terms of the qualities of a leader, such as honesty, consideration, trust, and fairness, as well as the position of a manager, based on communication, reinforcement, and role models [13]. Brown and Treviño conducted interviews with managers and ethics officers in corporate organizations and confirmed that ethical leadership has two dimensions: being a moral human and being a moral manager [32]. Furthermore, Heres and Lasthuizen, as well as later scholars, recognized these two sides as vital components of ethical leadership from both theoretical and empirical perspectives [23].

Leadership must influence followers [35]. Therefore, ethical leadership suggests that leaders influence members’ ethical behavior with modeling, such as learning using observation and imitation, from the perspective of the social learning theory. Bandura argued that “People learn by showing interest in role models to determine the appropriate behaviors, values, and attitudes to show [15].” Essentially, vicarious learning, in social learning theory, denotes the assimilation of desirable behaviors by examining the consequences and reward/penalty for behavior in the experiences of other people rather than personal ones [16].

### 2.3. Followership

Since the end of the 20th century, business organizations have deviated from preexisting methods to locate new forms of operations owing to changes in the economic environment, such as market liberalization, intensified competition among companies, and trade imbalances. Until recently, the leader’s role was overemphasized in facilitating the formation of a work environment for achieving high performance in organizations [36]. During this process, research on followership began emerging in earnest [37]. Leaders and followers play their respective roles, and the fact that anyone can be a leader and anyone can be a follower has gained prominence. This concept denotes that leaders exist because of their followers, and the value of leadership is recognized in followership [38]. Followers are those who provide initial assistance to support leaders, because they, in fact, perform tasks and achieve organizational success rather than remaining subordinate to leaders and managers [19].

Leadership and followership can be categorized into two types, according to their approach. The first refers to an examination of the correlation between followers from the perspective of leadership effectiveness, which is considered a lower concept of leadership. Wortman defined followership as attaining goals by participating in individual or collective efforts to achieve organizational goals under certain circumstances, as intended by the leader [39]. Conversely, this approach considers followership separately from leadership and regards it as an independent concept; this methodology emphasizes the importance of followers. Kelley stated that followers act regardless of their status and have the willpower to actively participate in the achievement of goals [19].

Uhl-Bien et al., presented a constructionist approach that examined the followership research published thus far, arguing that it is important to strengthen the status of leaders and followers together in the process of understanding leadership and followership [40]. First, the authors classified leadership studies into five types: leader-centric, follower-centric, relational view, role-based, and constructionist followership. Typical research theories in this category include leadership, identity, and process composition theories. Uhl-Bien et al., proposed a “reversed the lens” model that interprets leadership from the perspective of the followership process [40]. This model explains the followership process using the logic that the characteristics of followers influence behavior and the interaction process between the behaviors of leaders and followers forms the results of followership.

Despite various discussions in numerous studies on followership, engaging in a concrete and empirical approach is currently difficult. This study intends to use Kelley as the followership measurement tool, establishing the foundation for research on followership [19]. Classifying followership into two types, namely, independent critical thinking and active participation, is considered appropriate to the objectives of the current study.

### 2.4. Research Hypotheses

#### 2.4.1. Ethical Leadership and Followership

Research on ethical leadership and followership is mainly led by studies that analyze the impact of the relationship between leadership and followership on organizational performance. The current study classifies most studies in terms of the interaction process between leadership and followership: those that analyze the mediating effect of followership as an effect [41,42] and those that verify the mediating effect of followership on the relationship between transformational leadership and organizational performance [43]. Research on ethical leadership includes ethical managers’ abilities to respect independent critical thinking and encourage active participation using two-way communication with followers. Ethical leadership is essentially a social learning process using ethical leader behavior, namely, the role modeling and observational learning of followers. Ethical leadership is characterized by the quality of a leader, which, in turn, is characterized by honesty, consideration, trust, and fairness, and by the role of a manager that remains focused on communication, reinforcement, and being a role model. In this study, the quality aspect was similar to the independent critical disposition used by Kelley to classify followership.

Moreover, previous scholars have demonstrated aspects of a manager’s role in displaying similarities with the active participation of behavioral disposition [19]. Therefore, scholars have proposed that the higher the ethical leadership level is, the higher the level of followership with independent critical thinking and active participation is. Ideally, ethical leaders are honest, thoughtful, and fair in stimulating their followers’ independent critical thinking. Furthermore, they become active role models by transforming words into actions, instead of merely uttering words in front of their subordinates. Therefore, these efforts lead to a behavioral tendency toward active participation among followers.

Operating an ethical organization requires equal sacrifice and effort from leaders and followers. Followership is not simply about following a leader but also about identifying authentic leadership and making leaders great. Ethical leadership is based on social learning theory, and the influence of leadership is achieved with the followers’ observational learning of the role model characteristics exhibited by the leader. Accordingly, this study expects that the greater the influence of ethical leadership is, the better the followers perceive and learn.

Therefore, this study proposes the following hypothesis:

**Hypothesis** **1** **(H1):**
*Ethical leadership exerts a positive effect on followership.*


#### 2.4.2. Ethical Leadership and UPB

The relationship between ethical leadership and UPB can be observed using social learning theory. Building on this theory, employees can learn which behaviors are appropriate by observing and imitating the behaviors of their leaders [12]. Recent studies have established a negative relationship between ethical leadership and UPB, arguing that ethical leadership has the potential to avoid unethical behavior and imitate the behavior of ethical leaders [13,44,45,46,47,48,49]. Ethical leaders urge members to act ethically with rewards and punishments by setting an example of ethical behavior.

Accordingly, the current study presents the following hypothesis:

**Hypothesis** **2** **(H2):**
*Ethical leadership exerts a negative impact on UPB.*


#### 2.4.3. Followership and UPB

Domestic and international articles on followership and UPB (especially in the public sector) are difficult to find. However, this study predicts, with regard to the aforementioned concept of followership, that public officials display a tendency toward active participation (instead of independent critical thinking) to identify with the organization. This is because in relation to UPB, which is an individual-level variable, the followership of public officials seemingly exerts a greater effect on independent critical tendencies at the individual level instead of active participation in terms of organizational tendencies. This study expects that public officials who are independent and critical exert a certain degree of effect on restraining UPB.

In this regard, we provide the following hypothesis:

**Hypothesis** **3** **(H3):**
*Followership exerts a negative impact on UPB.*


#### 2.4.4. Mediating Effects of Followership

Studies have identified several possible mediators between ethical leadership and UPB. Increased moral attentiveness [47,50], affective organizational commitment [48], and responsibility taking [51] serve as mediators when ethical leadership lowers UPB. Moreover, studies have also found that ethical leadership fosters UPB with increased organizational trust and organizational identification [50], and reduced moral disengagement [47,50], thereby promoting identification with supervisors [51].

Few studies have directly analyzed the mediating effect of followership on the relationship between ethical leadership and UPB. However, several studies have examined the mediating effect of followership on the relationship between transformational and transactional leadership, and organizational effectiveness. We analyzed the mediating and regulatory effects of followership on organizational performance, including transformational and transactional leadership, job satisfaction, organizational engagement, organizational citizenship behavior, and innovation behavior [52,53]. Active participation and team spirit demonstrated mediating effects between job and organizational engagement. Jung proved that the mediating effects of followership, such as efficiency, effectiveness, and fairness, are significant between leadership and organizational performance [54]. Therefore, this study assumes that followership exerts a mediating effect on the relationship between ethical leadership and UPB.

Accordingly, we present the following hypothesis:

**Hypothesis** **4** **(H4):**
*Followership links ethical leadership and UPB.*


## 3. Research Design

### 3.1. Questionnaire Formulation

This study employed a questionnaire to examine the relationship between public officials’ ethical leadership in the central government and their subordinates’ followership and UPB. We adapted the questionnaire to measure ethical leadership, followership, and UPB to fit the public institution context. The questionnaire consisted of 9, 18, 6, 16, and 11 questions on ethical leadership (i.e., moral person and moral manager), followership (i.e., independent critical thinking and active participation), UPB, organizational culture, and the demographic variables of the direct superior and research subject, respectively. Additional information is available in Appendix A.

#### 3.1.1. Ethical Leadership

As a tool for measuring ethical leadership, the questionnaire was created after translating the Ethical Leadership Scale, which was developed by Brown et al. [13]. The questionnaire consists of nine items rated on a 5-point Likert-type scale (1 = not at all; 5 = strongly agree).

#### 3.1.2. Followership

Kelley isolated the relationship between leadership and followers and emphasized the role of followers as an independent element instead of a sub-element of leadership [19]. Two elements were suggested as characteristics of followers: cognitive disposition (independent and critical thinking) and behavioral disposition (active participation). The survey presented 18 items that included independent, critical thinking, and active participation, rated on a 5-point Likert-type scale ranging from 1 (not at all) to 5 (very much).

#### 3.1.3. UPB

The South Korean translation of Lee and Jeon (2016) was used for the survey questionnaire by Umphress et al., to measure UPB [7,55]. However, the content of the survey is related to the company; thus, it is difficult to apply it to public organizations in its original format. Therefore, the survey content was revised to suit the management of civil services. The questionnaire consisted of six items rated on a 5-point Likert-type scale related to unethical and pro-organizational behavior (1 = not at all; 5 = strongly agree).

#### 3.1.4. Organizational Culture

Organizational culture was used as the control variable. It is an important element of organizational behavior and pertains to a set of shared values and norms that influence the attitudes and behaviors of individuals, groups, and organizations. Cameron and Quinn classified relationship- (group culture), innovation- (development culture), hierarchy- (hierarchical culture), and task-oriented (rational culture) cultures according to the competing values model [56]. To measure organizational culture, we used the questionnaire developed by Cameron and Quinn [56]. Sixteen questions were used to measure this construct on a 5-point Likert-type scale (1 = not at all; 5 = strongly agree).

### 3.2. Demographic Variables

This study investigated the relationship among ethical leadership, followership, and the UPB of public officials working in central departments using six questions on gender, age, employment period, position, career path, educational background, and affiliated institutions. Gender was dichotomized into male (1) and female (2), and age was measured in the following order: below 30 (1), 30–39 (2), 40–49 (3), and over 50 (4). Employment periods were divided as follows: 5 years or less (1), 6–10 years (2), 11–15 years (3), 16–20 years (4), 21–25 years (5), and more than 26 years (6). The positions were Grades 4 (1), 5 (2), 6 and 7 (3), and others (4). Career paths denote open recruitment at Grades 5 (1), 7 (2), 9 (3), and others (4). Educational background was classified as follows: lower than high school education (1), diploma holder (2), bachelor’s degree holder (3), and master’s degree holder (4). The affiliated institutions were as follows: Ministry of Economy and Finance (1); Ministry of Education (2); Ministry of Foreign Affairs (3); Ministry of Unification (4); Ministry of Culture, Sports, and Tourism (5); Ministry of Health and Welfare (6); Ministry of Environment (7); Ministry of Employment and Labor (8); Anti-Corruption & Civil Rights Commission (9); and National Statistical Office (10).

The superiors of the participants were measured using five items: gender, age, length of employment, position, and period worked together. Sex was classified as male (1) or female (2), and age was measured as follows: below 30 (1), 30–39 (2), 40–49 (3), and over 50 (4). The employment periods were 5 years or less (1), 6–10 years (2), 11–15 years (3), 16–20 years (4), 21–25 years (5), and more than 26 years (6). The positions were section chief (1), general manager (2), head of department (3), and others (4). The duration worked together was measured separately.

### 3.3. Data Collection and Analysis Methods

#### 3.3.1. Data Collection

First, the study used stratified sampling, a type of probability sampling, after considering the sampling costs and time constraints for survey target identification. Korea’s national administrative organization currently comprises 22 ministries, 18 offices, and 7 committees. The sample was selected according to regional factors. Ten organizations were selected for this study (the selected organizations were Ministry of Economy and Finance; Ministry of Education; Ministry of Foreign Affairs; Ministry of Unification; Ministry of Culture, Sports, and Tourism; Ministry of Health and Welfare; Ministry of Environment; Ministry of Employment and Labor; Anti-corruption & Civil Rights Commission; and National Statistical Office). When collecting data, divisions were selected for each department or bureau by using systematic sampling, and each division was selected at intervals with continuous rules. Adjacent divisions were added if the selected divisions required more research samples.

The unit of analysis was selected from public officials with Grade 4 or lower working in 10 central government ministries, and the questionnaire was used as the survey method. The empirical analysis section presented the objectives of the study, and the questionnaires were distributed for empirical analysis. Ten questionnaire collectors were selected and trained in advance for the survey. Finally, the data were collected. The cross-sectional survey was conducted for three weeks, beginning 7 August 2017. Overall, 408 questionnaires were collected, and data from 404 questionnaires were analyzed, excluding 4 with missing data on essential items.

Ethical clearance for the study was sought from the committee of the university where the first author was employed. All ethical considerations for the research involving human subjects, including informed consent and anonymity, were upheld. Participation in the survey was voluntary. Survey participants could leave any questions they did not wish to answer unanswered. They were also allowed to refuse to participate or withdraw at any time. Their answers remained anonymous and confidential.

#### 3.3.2. Method of Analysis

This study applied various statistical methods to analyze the impact of ethical leadership on followership and UPB. Additionally, organizational culture was set and analyzed as a control variable, along with demographic variables. First, to examine the overall reliability and validity of the measurement, the study conducted a basic statistical analysis of the respondent status, followed by reliability and factor analyses of the measurement tool. Second, the study analyzed all variables’ mean, standard deviation, and minimum and maximum values for descriptive analysis and performed relevance verification and correlation analyses, such as mean difference analysis. Third, in relation to the proof of the hypotheses that aimed to examine the effect of ethical leadership on followership, the effect of followership on UPB, and the mediating effect of followership, multiple regression and Process Macro by Hayes were used as analysis tools [57].

Multiple regression analysis was used to determine the relationship between independent and dependent variables and to test Hypotheses 1 to 3. Process Macro was used to test Hypothesis 4. The collected data were analyzed using SPSS 26.

In Process Macro, the bootstrap method was applied to verify the mediating effect. The number of default samples was set to thousands, and the analysis results were displayed after the relevant variables were inputted. This method has the advantage of avoiding numerous procedures, such as the Baron and Kenny method or Sobel test [58,59].

As the data were collected from self-reported questionnaires measured using a single source, concerns about common method bias may have been raised. We conducted Harman’s single-factor test on the main variables (dependent, independent, and mediating) included in the research model. The explanatory variance of a single factor was 11.24%, which is far below 50%. The results indicated that severe common method bias was not present in this study.

## 4. Research Results

### 4.1. Reliability and Validity of the Measurements

#### 4.1.1. Sample Characteristics

Table 1 displays the questionnaire data used for the final analysis and summarizes the following items based on 404 questionnaires, including 44 from the National Statistical Office and 40 copies from the other nine ministries: the gender, age, employment period, position, career path, and educational background of the respondents, and the gender, age, employment period, position, and period worked together with their superiors. The composition of the participants was arranged to be 54.8% (male) and 45.2% (female) to balance the number of government officials in terms of gender.

As for age, 39.1% and 37.6% were in their 30s and 40s, respectively, accounting for 76.7% of total respondents. Additionally, 14.8% were above 50 years of age, and 8.5% were above 20 years of age. In terms of service period, those with less than 5 years, 6–10 years, and 11–15 years accounted for 25.6%, 20.6%, and 20.1%, respectively. Finally, those aged below 15, 16–20, 21–25, and more than 26 accounted for 66.3%, 10.4%, 16.5%, and 6.9%, respectively. Regarding position, Grades 5 and 6, and Grade 7 reached 47.7% and 37.9%, respectively, accounting for 85.6% of the total. Grade 4 and other grades accounted for 6.4% and 8.0% of cases, respectively. Regarding the career path, open recruitment at Grades 5, 7, 9, and others represented 25.3%, 32.0%, 23.8%, and 18.9%, respectively, indicating an even distribution across grades. Regarding educational background, 71.8% and 23.1% of the respondents were university graduates and postgraduates, respectively, accounting for 94.9% of the total sample. College graduates accounted for 3.5% of the sample, and high school graduates or below constituted 1.6%.

Regarding the subjects’ superiors, 79.8% were male, and 20.2%, female. In terms of age, 62.2% and 33.1% were in their 40s and 50s, respectively, accounting for 95.3% of the total. The remaining 4.7% were superiors in their 30s. The employment periods were 5 years or lower (2.1%), 5–10 years (2.1%), 11–15 years (24.8%), 16–20 years (33.9%), 21–25 years (25.9%), and 26 years or more (10.7%). This study found that direct superiors were evenly distributed among middle-grade executives. The majority of the positions included chief managers (82.2%), others (8.7%), general managers (7.9%), and heads of department (1.3%). Those who worked together for less than one year accounted for 61.1%, and those who worked for one or two years together accounted for 31.9%, constituting a total of 93.0%. Finally, the remaining 4.3% and 2.7% pertained to work periods of 2–3 years and more than 3 years, respectively.

#### 4.1.2. Reliability Verification

Reliability analysis was conducted to examine whether the concepts were consistently measured by the respondents. This study aimed to verify the reliability of the measurement tool using Cronbach’s alpha coefficient. Specifically, the objective of this step was to obtain a reliability coefficient that indicates the degree of reliability, given that a positive correlation between responses exists. Responses to each item are based on the fact that they demonstrate regular patterns. Therefore, the alpha coefficient serves as a standard to explain the cohesion among the variables used in the reliability analysis. When the coefficient is 0.30 or lower, the cohesion between items is weak; conversely, when it is 0.70 or more, the cohesion between items is strong. Based on these standards, this study determined that if the alpha coefficient was greater than or equal to 0.70, the measured items implied internal consistency (Table 2).

The reliability of the tool was analyzed using Cronbach’s alpha, which measures intra-item consistency. The results indicate that all the variables displayed high-reliability coefficient values of 0.80 or more. The reliability values for ethical leadership, followership, and UPB were 0.951, 0.946, and 0.831, respectively, higher than 0.70, indicating reliability [60].

The method for testing normal distribution was tested for skewness, which represents the degree of bias, and kurtosis, which indicates the degree of sharpness. A normal distribution can be assumed when the absolute values of skewness and kurtosis are less than 3. The absolute values of skewness and kurtosis of the tool were less than 3; thus, normality could be assumed.

#### 4.1.3. Validity Verification

To verify the validity of the measurements, this study conducted a confirmatory factor analysis. The goodness of fit of the measurement model was confirmed using the value and goodness-of-fit indices. This value is sensitive to sample size; thus, the null hypothesis can be easily rejected. Therefore, when evaluating fit, studies should rely not only on verification but also on other fit indices [61]. The values were checked, where the goodness-of-fit index (GFI), comparative fit index (CFI), normed fit index (NFI), Tucker–Lewis Index (TLI), and root mean square error of approximation (RMSEA) values were used as the model evaluation criteria. If the standards of the general goodness-of-fit index for adopting the model are 0.90 or more for GFI, CFI, NFI, and TLI, and 0.05 or less for RMSEA, then they are considered the best fit. However, if their values are less than 0.08, they are considered a good fit. Values less than 0.10 are considered a normal fit.

The results of the analysis of the measurement model in Table 3 indicate that the standard value to be evaluated as a good fit was fulfilled and the factor composition was deemed appropriate.

As seen in Table 4, ethical leadership consisted of nine measurement variables with factor loadings of 0.686–0.932, which exceed 0.50. The CR value was greater than 1.96, which is statistically significant (*p* < 0.001) and indicates that the measurement model was suitable. Furthermore, followership displayed factor loadings of 0.523 to 0.802, which exceed 0.50. UPB was also significant between 0.556 and 0.874. Therefore, the measurement model was appropriate. Confirmatory factor analysis revealed that the measurement tools for ethical leadership, followership, and UPB were valid.

### 4.2. Descriptive Statistics and Correlation Analysis

This study analyzed the survey data and presented descriptive statistics, such as mean and standard deviation, for the sum of the items on ethical leadership, followership, and UPB, which were rated using a Likert-type scale.

Table 2 shows that the averages of the survey were 3.96 for ethical leadership, 3.49 for followership, and 3.06 for UPB. In summary, the UPB score was the lowest. Alternatively, it was possible to confirm the degree of distribution using standard deviation and range (maximum–minimum values). The means of the variable were 3.96 for ethical leadership, 3.49 for followership, and 3.06 for UPB. Furthermore, the average deviation between ethical leadership and UPB was the largest.

The results of the correlation analysis among variables were considered for hypothesis testing. First, a significant positive (+) correlation was observed between ethical leadership and followership (*r* = 0.473, *p* < 0.001). This finding was consistent with our hypothesis. Furthermore, no correlation was observed between followership and UPB.

### 4.3. Hypothesis Verification

The study set the relationships between ethical leadership and followership, and between followership and UPB as the hypotheses. To verify this, correlation and multiple regression analyses were conducted. Process Macro was used to test the hypothesis of the mediating effect of followership on the relationship between ethical leadership and UPB.

#### 4.3.1. Verification Results for Hypothesis 1

To perform a regression analysis of the effect of ethical leadership on followership, this study reviewed the autocorrelation of the dependent variable and multicollinearity between the independent variables. The Durbin–Watson index was 1.906. No multicollinearity was observed, because the variance inflation factor (VIF) values for understanding multicollinearity between the variables were 1.374–3.809, which are less than 10. Therefore, this study assumed that the data were appropriate for regression analysis. Hypothesis 1 indicated that ethical leadership (*p* < 0.001) exerted a significant positive (+) effect on followership as a result of the regression analysis (Table 5). In addition, the higher the level of ethical leadership (=0.278) was, the greater the level of followership was. The explanatory power to explain followership was 34.5%.

#### 4.3.2. Verification of Ethical Leadership, Followership, and UPB (Hypotheses 2 and 3)

Table 6 displays the regression analysis results of the effects of ethical leadership and followership on UPB. The Durbin–Watson index was 1.983, indicating independence without autocorrelation. Ethical leadership (Hypothesis 2) did not exhibit a significant effect, whereas followership (*p* = 0.004 < 0.01) exerted a significant effect on UPB (Hypothesis 3). The lower the followership level (B = −0.258) was, the higher the UPB level was, with an explanatory ability of 16.3%.

#### 4.3.3. Verification of Ethical Leadership, Followership, and UPB (Hypothesis 4)

As shown in Table 7, the mediating effect of followership was tested using a path analysis to determine the effect of ethical leadership on UPB. The results indicate that the mediating effect of followership (B = −0.059, *p* < 0.05) was significant (Hypothesis 4). Moreover, the findings indicated that the higher the level of ethical leadership was, the higher the level of followership was, resulting in lower levels of UPB. Followership demonstrated a direct mediating effect on leadership and UPB.

## 5. Discussion

### 5.1. Theoretical Implications

Our research provides theoretical insights into the leadership, followership, and UPB literature. First, the positive relationship between superiors’ ethical leadership and followership suggests that the social learning theory, which forms the theoretical basis of ethical leadership, improves followership among subordinates. These results indicate the influence of superiors’ ethical leadership, active learning, and acceptance of ethical values via followership. The completion of the leadership process relies on appropriate followers’ responses. This study suggests that the key elements in high followership, such as independence, critical thinking, and active engagement, can lead to the acceptance and completion of the ethical leadership process.

Second, the results regarding followership and unethical pro-organizational behavior (UPB) indicate that followers’ characteristics and attitudes negatively impact UPB. The independent and critical thinking of followership theory enables an accurate perception and cognition of pro-organizational and unethical behaviors to be achieved. Active engagement translates these judgments into proactive actions. This result aligns with recent studies showing a relationship between organizational identification, employees’ assimilation of organizational values, and high UPB [62,63,64,65]. These findings underscore the importance of followership in overcoming the limitations of employees’ inability to make independent judgments.

Third, this study hypothesizes that ethical leadership negatively affects ethical leadership on UPB. However, no significant differences were observed between the groups. Although many studies have demonstrated a negative relationship between ethical leadership and UPB, others have indicated a more complex relationship between these two variables [50,51,66,67]. In other words, the learning mechanism of ethical leadership may not directly influence followers’ behavior. This result suggests that other variables may be involved in the relationship between these two variables.

Fourth, research findings showing a full mediation effect of followership indicate that followership is necessary for ethical leadership to influence UPB. Followership is based on a followership-centered perspective that completes the leadership process; ethical leadership within an organization is achieved with role modeling, reinforcement of ethical leadership by superiors, and active learning via subordinates’ followership. Therefore, a reduction in UPB can be achieved.

### 5.2. Practical Implications

The results have several practical implications. First, the research findings reveal that superiors’ ethical leadership positively impacts subordinates’ followership. Therefore, public organizations must enhance ethical leadership abilities among superiors and provide programs and education emphasizing ethical values, enabling them to demonstrate morally upright behavior. Additionally, followership among subordinates is closely related to organizational leadership and can play a crucial role in preventing UPB. Hence, public organizations should introduce followership development programs to improve subordinates’ independent and critical thinking, active and proactive engagement, and ethical consciousness.

Second, the relationship between followership and UPB suggests that if followers critically consider organizational values and ethical principles using independent thinking and actively pursue the public organization’s interests and ethical behavior, they can prevent UPB. To achieve this, organizations must cultivate an ethical environment. They should emphasize ethical norms and values, encourage ethical behavior, and implement measures, such as reward systems, to raise awareness of ethical conduct.

Finally, public organizations can improve their internal culture to prevent UPB. It is important to establish an ethical value system and ensure the proper implementation of regulations and rules within the organization to uphold these values. Precisely, leaders must adjust their leadership styles to support their followers’ independent, critical thinking, and active engagement. In addition, leaders should encourage individuals to express their opinions, promote participation, respect diversity, and foster open communication.

## 6. Limitations and Avenues for Future Research

This study has the following limitations and offers suggestions for future studies. First, it did not examine the moderating effect on the relationship between ethical leadership and followership. Various variables, such as leader trust, commitment, and organizational identification, can be considered moderating variables affecting followership. In particular, explanatory variables, such as organizational identification, must also be considered in the context of the relationship among followership, mediator, and UPB.

Second, this study uses organizational culture as a control variable. However, from the perspective of organizational culture, it is possible to examine how the dependent variables are related to the relationship between ethical leadership and followership, according to the characteristics of each department. In addition, considering various aspects of the variables that appear by comparing these effects with those of private organizations, it will be possible to develop a more detailed study.

Third, although we contextually modified the questionnaire to measure ethical leadership, followership, and UPB for public organizations, it may not match the organizational behavior applied to public institutions [7,11]. The items need to be adjusted concerning the occupational characteristics of the subjects [68]; therefore, research reflecting the nature of the occupational group is required.

Fourth, it is necessary to generalize the results to the public sector by broadening the scope of the research. For instance, due to the influence of the leading factors of institutions, community systems, and culture, the research results of local governments can be different. Therefore, it is necessary to review the differences and similarities with respect to the results of this study by researching local governments.

## 7. Conclusions

Our study emphasizes the critical role of followership in fostering UPB by providing superior ethical leadership. Furthermore, it highlights the role of followership in the process by which leaders foster ethical behavior, given that the dynamics increase subordinates’ followership and decrease their UPB. This study is one of the first to provide empirical evidence of how ethical leadership facilitates employees’ UPB focusing on the mediating role of subordinates’ followership.

## Figures and Tables

**Figure 1 behavsci-13-00454-f001:**
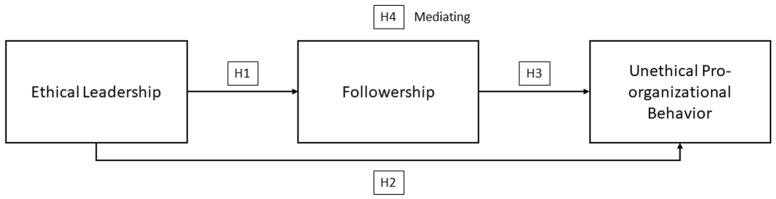
Research Model.

**Table 1 behavsci-13-00454-t001:** Participant characteristics.

	Variable		N	%
Oneself	Gender	Men	213	54.8
		Women	176	45.2
	Age	20s	34	8.5
		30s	156	39.1
		40s	150	37.6
		Over 50 years old	59	14.8
	Employment period	5 years or less	101	25.6
		6–10 years	81	20.6
		11–15 years	79	20.1
		16–20 years	41	10.4
		21–25 years	65	16.5
		More than 26 years	27	6.9
	Position	Grade 4	25	6.4
		Grade 5	185	47.7
		Grades 6 and 7	147	37.9
		Other	31	8.0
	Career path	Open recruitment at Grade 5	98	25.3
		Open recruitment at Grade 7	124	32.0
		Open recruitment at Grade 9	92	23.8
		Other	73	18.9
	Educational background	High school or below	6	1.6
		Diploma holder	13	3.5
		Bachelor’s degree holder	270	71.8
		Over master’s degree level	87	23.1
Superior	Gender	Men	308	79.8
		Women	78	20.2
	Age	20s		
		30s	18	4.7
		40s	239	62.2
		More than 50 years	127	33.1
	Years of employment	5 years or less	8	2.1
		6–10 years	10	2.7
		11–15 years	93	24.8
		16–20 years	127	33.9
		21–25 years	97	25.9
		More than 26 years	40	10.7
	Position	Chief manager	313	82.2
		General manager	30	7.9
		Department head	5	1.3
		Other	33	8.7
	Period worked together	Less than 1 year	226	61.1
		1–2 years	118	31.9
		2–3 years	16	4.3
		3 years or more	10	2.7

**Table 2 behavsci-13-00454-t002:** Reliability verification results.

Variable	N	Minimum Value	Maximum Value	Average	Standard Deviation	Skewness	Kurtosis	Cronbach’s α
Ethical leadership	404	1.22	5.00	3.96	0.79	−0.757	0.649	0.951
Followership	404	1.50	5.00	3.49	0.60	0.345	0.553	0.946
UPB	404	1.00	5.00	3.06	0.69	−0.087	0.897	0.831

**Table 3 behavsci-13-00454-t003:** Factor analysis (1).

	χ^〖2〗	df	p	GFI	CFI	NFI	TLI	RMSEA
Ethical leadership	127.238	25	<0.001	0.931	0.968	0.961	0.954	0.093
Followership	241.481	64	<0.001	0.910	0.937	0.917	0.924	0.085
UPB	23.286	3	<0.001	0.977	0.976	0.973	0.919	0.033
Criteria			>0.05	>0.90	>0.90	>0.90	>0.90	<0.10

Note: Goodness-of-fit index (GFI), comparative fit index (CFI), normed fit index (NFI), Tucker–Lewis Index (TLI), and root mean square error of approximation (RMSEA).

**Table 4 behavsci-13-00454-t004:** Confirmatory factor analysis (2).

		B	SE	β	CR	p	SMC
Ethical leadership	a1	1.000		0.843			0.771
	a2	1.012	0.037	0.866	27.528	<0.001	0.749
	a3	0.889	0.058	0.686	15.420	<0.001	0.470
	a4	0.893	0.048	0.776	18.442	<0.001	0.602
	a5	0.889	0.044	0.825	20.327	<0.001	0.680
	a6	1.040	0.045	0.889	23.133	<0.001	0.790
	a7	1.134	0.045	0.932	25.298	<0.001	0.869
	a8	0.841	0.051	0.725	16.654	<0.001	0.526
	a9	1.035	0.048	0.860	21.788	<0.001	0.739
Followership	b2	1.000		0.653			0.396
	b3	1.129	0.083	0.802	13.630	<0.001	0.364
	b4	1.105	0.083	0.778	13.298	<0.001	0.450
	b6	1.087	0.081	0.783	13.362	<0.001	0.273
	b7	0.961	0.078	0.706	12.264	<0.001	0.441
	b8	1.075	0.081	0.774	13.239	<0.001	0.457
	b9	1.084	0.084	0.752	12.927	<0.001	0.565
	b11	0.909	0.077	0.676	11.825	<0.001	0.599
	b13	1.038	0.089	0.664	11.652	<0.001	0.498
	b15	0.663	0.070	0.523	9.424	<0.001	0.613
	b16	0.944	0.080	0.671	11.746	<0.001	0.605
	b17	0.875	0.082	0.603	10.714	<0.001	0.643
	b18	0.990	0.089	0.629	11.114	<0.001	0.426
UPB	c1	1.000		0.800			0.641
	c2	1.122	0.073	0.874	15.388	<0.001	0.764
	c3	0.797	0.067	0.617	11.837	<0.001	0.381
	c5	0.764	0.069	0.581	11.084	<0.001	0.338
	c6	0.710	0.068	0.556	10.509	<0.001	0.309

Note: B (unstandardized β), SE (standard error), β (standardized β), CR (concept reliability), *p* (significance probability), and SMC (squared multiple correlation).

**Table 5 behavsci-13-00454-t005:** Impact of ethical leadership on followership.

Variable	B	SE	β	t	p
Constant	1.056	0.436		2.421	0.016
Gender (female)	−0.061	0.069	−0.050	−0.887	0.376
Age	0.006	0.071	0.007	0.081	0.936
Employment period	−0.017	0.038	−0.043	−0.461	0.645
Position	−0.099	0.057	−0.118	−1.730	0.085
Career path	0.076	0.041	0.136	1.861	0.064
Educational background	0.176	0.063	0.153	2.813	0.005
Superior’s gender (female)	−0.058	0.087	−0.037	−0.666	0.506
Superior’s age	−0.115	0.079	−0.103	−1.464	0.145
The period of employment of superiors	0.116	0.042	0.203	2.799	0.006
Superior’s position	−0.040	0.036	−0.059	−1.124	0.262
Period worked together	0.001	0.052	0.001	0.020	0.984
Community culture	0.042	0.067	0.057	0.629	0.530
Organism culture	0.180	0.073	0.248	2.472	0.014
Hierarchical culture	0.117	0.054	0.127	2.177	0.030
Market culture	0.005	0.066	0.006	0.077	0.939
Ethical leadership	0.228	0.051	0.278	4.453	0.000
	Adj R^〖2〗 = 0.345 F = 9.281 p = 0.000				

Note: B (unstandardized coefficient); SE (standard error); β (standardized coefficient); *p* (*p*-value).

**Table 6 behavsci-13-00454-t006:** Effects of ethical leadership and followership on UPB.

Variable	B	SE	β	t	p
Constants	4.066	0.622		6.541	0.000
Gender (female)	−0.104	0.094	−0.070	−1.101	0.272
Age	0.130	0.097	0.138	1.333	0.184
Employment period	−0.099	0.052	−0.202	−1.915	0.057
Position	−0.210	0.079	−0.206	−2.668	0.008
Career path	0.121	0.056	0.179	2.158	0.032
Educational background	0.164	0.087	0.118	1.878	0.062
Superior’s gender (female)	−0.108	0.119	−0.057	−0.905	0.366
Superior’s age	0.028	0.108	0.021	0.262	0.794
Period of employment of superiors	0.025	0.058	0.036	0.429	0.668
Superior’s position	0.016	0.049	0.020	0.326	0.745
Period worked together	−0.027	0.072	−0.024	−0.376	0.707
Community culture	0.113	0.092	0.126	1.233	0.219
Organism culture	−0.094	0.101	−0.106	−0.927	0.355
Hierarchical culture	−0.108	0.074	−0.097	−1.450	0.148
Market culture	0.198	0.091	0.194	2.176	0.031
Ethical leadership	0.132	0.075	0.134	1.764	0.079
Followership	−0.258	0.090	−0.213	−2.871	0.004
	Adj R^〖2〗 = 0.163 F = 3.736 p = 0.000				

Note: B (unstandardized coefficient); SE (standard error); β (standardized coefficient); *p* (*p*-value).

**Table 7 behavsci-13-00454-t007:** Mediating effect of followership.

							B	SE	95% CI	
									LLCI	ULCI
Ethical leadership	→	Followership			→	UPB	−0.059	0.028	−0.120	−0.014

Note: B (unstandardized coefficient); SE (standard error); LLCI (lower-limit confidence interval); ULCI (upper-limit confidence interval).

## Data Availability

The participants of this study did not give written consent for their data to be shared publicly, so due to the sensitive nature of the research, supporting data are not available.

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
