# Peer review of "Impact of Superiors’ Ethical Leadership on Subordinates’ Unethical Pro-Organizational Behavior: Mediating Effects of Followership"

_behavsci, 2023, doi:10.3390/bs13060454_

Round 1
Reviewer 1 Report
Dear authors,
I find your paper very interesting and important for public sector organizations. This study confirmed that followership had a significant influence on UPB, and suggested ethical leadership was an important precedent factor of followership. You presented your methodology in a proper way, as results. Also, I find your implications suitable for publication, too. You even proved CMB by using Harmans factor, which is important.
Author Response
We would like to express our appreciation to the peer reviewer who provided thoughtful and scholarly evaluations of our manuscript.
Reviewer 2 Report
Dear author(s)
It was my pleasure to review your manuscript entitled “Impact of Superiors’ Ethical Leadership on Subordinates’ Unethical
Pro-Organizational Behavior: Mediating Effects of Followership” and advise you to prosper your current research project. In my view, your topic has touched on a critical issue in a fascinating context. However, there are many spaces to be improved in terms of argumentation, theoretical background, research method, and findings. I hope my below comments would help you develop your work into groundbreaking research in your domain.
In the abstract, the conclusion section, the authors should write the conclusion of the article.
Since this article is going to be published in 2023, more recent sources should be used in the theoretical literature and introduction. Maximum 10 references.
Much sharper problematization is required so that the introduction draws the reader into the paper. The introduction therefore needs to do a better job in setting the stage for the articulation of the theoretical contributions of the study. At the end of the introduction, we should have a clear idea of what the paper is about (i.e. its motivation, the gap in understanding that the paper is trying to address and summary of theoretical contributions).
The authors are better to explain the gap in this article further.
Theoretical literature has not been considered and reviewed. It’s better to observe the connection between the contents. Try to explain everything except the topics in order to establish the necessary
coherence.
What are the theoretical and practical implications of your study and which limitations and possible future research emerge from it? At the moment. the chapter is that is now entitled as "Conclusion" should link back to the literature and show theoretical contributions, that exceed the conclusion that some literature was "inline" with the findings of the authors.
What scale has been used for analysis? What are the results of your research and how can it help your statistical community?
The authors need to draw substantive conclusions from their results, and suggest and develop more recommendations for further research.
Add limitation.
- Using the following references could be beneficial as these add more evidence to the literature review section:
The effect of team performance on the internationalization of Digital Startups: the mediating role of entrepreneurship. International Journal of Human Capital in Urban Management, 8(1), 17-30. doi: 10.22034/IJHCUM.2023.01.02
Investigating social capital, trust and commitment in family business: case of media firms. Journal of Family Business Management. https://doi.org/10.1108/JFBM-02-2021-0013
Best of luck with the further development of the paper.
Another round of spellchecking by a native speaker is recommended.
Author Response
Response to the Reviewer 2
We would like to express our appreciation to the peer reviewer who provided thoughtful and scholarly evaluations of our manuscript. We tried to apply all your suggestions. We marked the revised part of the manuscript in green.
In the abstract, the conclusion section, the authors should write the conclusion of the article.
- We added in the abstract, " The study concludes with the theoretical and practical implications of these findings, along with the limitations of the study."
Since this article is going to be published in 2023, more recent sources should be used in the theoretical literature and introduction. Maximum 10 references.
- We added the following 16 relevant references, including 11 articles published since 2019.
- Lord, R.G., Epitropaki, O., Foti, R.J., Hansbrough, T.K. Implicit leadership theories, implicit followership theories, and dynamic processing of leadership information. Annual Review of Organisation Psychology and Organizational Behavior. 2020. 1, 49-74. https://doi.org/10.1146/annurev-orgpsych-012119-045434
- DeRue, D.S., Ashford, S.J. Who will lead and who will follow? A social process of leader identity construction in organizations. Academy of Management Review. 2010. 35:627–47. https://doi.org/10.5465/AMR.2010.53503267
- Steffens, N.K., Munt, K.A., van Knippenberg, D., Platow, M.J., Haslam, S.A. Advancing the social identity theory of leadership: a meta-analytic review of leader group prototypicality. Organizational Psychology Review. 2021. 11, 35-72, doi: 10.1177%2F2041386620962569.
- Carsten, M.K., Uhl-Bien, M. Ethical followership: an examination of followership beliefs and crimes of obedience. Journal of Leadership and Organizational Studies. 2013.20. 49-61, doi: 10.1177/1548051812465890.
- Lu, CS., Lin, CC. The Effects of Ethical Leadership and Ethical Climate on Employee Ethical Behavior in the International Port Context. J Bus Ethics. 2014. 124, 209–223. https://doi.org/10.1007/s10551-013-1868-y
- Pio, R.J., Lengkong, F.D.J. The relationship between spiritual leadership to quality of work life and ethical behavior and its implication to increasing the organizational citizenship behavior. Journal of Management Development. 2020. 39, 293-305. https://doi.org/10.1108/JMD-07-2018-0186
- Tajpour, M., Razavi, S. The effect of team performance on the internationalization of Digital Startups: the mediating role of entrepreneurship. International Journal of Human Capital in Urban Management.2023. 8, 17-30. doi: 10.22034/IJHCUM.2023.01.02
- Tajpour, M., Salamzadeh, A., Salamzadeh, Y., Braga, V. Investigating social capital, trust and commitment in family business: case of media firms. Journal of Family Business Management. 2020. 12, 938-958. https://doi.org/10.1108/JFBM-02-2021-0013
- Alniacik, E., Kelebek, E. F. E., Alniacik, U. The moderating role of message framing on the links between organizational identification and unethical pro-organizational behavior. Management Research Review. 2021. 45, 502-523. https://doi.org/10.1108/MRR-01-2021-0004
- Baur, C., Soucek, R., Kuhnen, U., Baumeister, P. F. Unable to resist the temptation to tell the truth or to lie for the organization? Identification makes the difference. Journal of Business Ethics. 2020. 167, 643–662. https://doi.org/10.1007/s10551-019-04162-3
- Irshad, M., Bashir, S. The dark side of organizational identification: A multi-study investigation of negative outcomes. Frontiers in Psychology. 2020. 11, 2521. https://doi.org/10.3389/fpsyg.2020.572478
- Kim, M. Y., Miao, Q., Park, S. M. Exploring the relationship between ethical climate and behavioral outcomes in the Chinese public sector: The mediating roles of affective and cognitive responses. International Journal of Business, Humanities and Technology. 2015. 5, 88–103.
- Miao, Q., Newman, A., Yu, J., Xu, L. The relationship between ethical leadership and unethical pro-organizational behavior: Linear or curvilinear effects?. Journal of Business Ethics. 2013. 116, 641–653. https://doi.org/10.1007/s10551-012-1504-2
- Naseer, S., Bouckenooghe, D., Syed, F., Khan, A. K., Qazi, S. The malevolent side of organizational identification: Unraveling the impact of psychological entitlement and manipulative personality on unethical work behaviors. Journal of Business and Psychology. 2020. 35, 333–346. https://doi.org/10.1007/s10869-019-09623-0
- Wang, Y. J., Li, H. Moral leadership and unethical proorganizational behavior: A moderated mediation model. Frontiers in Psychology. 2019. 10, 1–19. https://doi.org/10.3389/fpsyg.2019.02640
- Zhang, X. C., Yao, Z. Impact of relational leadership on employees' unethical pro-organizational behavior: A survey based ontourism companies in four countries. PLoS ONE. 2019. 14(12), 1–19. https://doi.org/10.1371/journal.pone.0225706
Much sharper problematization is required so that the introduction draws the reader into the paper. The introduction therefore needs to do a better job in setting the stage for the articulation of the theoretical contributions of the study. At the end of the introduction, we should have a clear idea of what the paper is about (i.e. its motivation, the gap in understanding that the paper is trying to address and summary of theoretical contributions). The authors are better to explain the gap in this article further.
- We have summarized the theoretical contributions of this paper as follows.
“The findings of this study contribute to the literature on leadership and organizational ethics in multiple ways. First, it contributes to existing research on ethical leadership by introducing the concept of followership and examining its role in promoting ethical consciousness. Identifying the mutual influence between leaders and followers, especially the role of followers in promoting ethical conduct within an organization, is a crucial gap in this field [20–21].
Second, this study was among the first to investigate the mediating role of followership in leadership and UPB. Contrary to the traditional leader-centric approach in the leadership literature, this study takes a role-based followership approach and examines the development of organizational ethics by considering subordinates and their characteristics.
Third, this empirical study was conducted in a public organizational setting. Existing research on ethical consciousness has mostly focused on the corporate sector, and detailed empirical analysis of ethical consciousness in the public sector is needed [23 – 25]. It is essential to acknowledge the differences in characteristics between public and private organizations and to examine them separately. Through a critical analysis of ethical leadership and unethical behavior, this study aims to expand the scope of research beyond corporate-oriented research and provide valuable insights for promoting ethical behavior in the public sector.”
Theoretical literature has not been considered and reviewed. It’s better to observe the connection between the contents. Try to explain everything except the topics in order to establish the necessary coherence.
- Since the leading theory we employed is ethical leadership based on social learning theory, we reviewed more theoretical literature in this article.
What are the theoretical and practical implications of your study and which limitations and possible future research emerge from it? At the moment. the chapter is that is now entitled as "Conclusion" should link back to the literature and show theoretical contributions, that exceed the conclusion that some literature was "inline" with the findings of the authors.
- We have divided the current implications into theoretical and practical parts and have significantly reinforced related content.
What scale has been used for analysis? What are the results of your research and how can it help your statistical community?
- Scales were added as a supplementary material. The questionnaire items modified to suit the context of public organizations will be helpful to our statistical community as they have great potential to be used for quantitative analysis in other studies in the future.
The authors need to draw substantive conclusions from their results, and suggest and develop more recommendations for further research.
Add limitation.
- We divided the current conclusion into sections into theoretical/practical implications, limitations, future suggestions, and conclusions and greatly enriched the contents.
Using the following references could be beneficial as these add more evidence to the literature review section:
The effect of team performance on the internationalization of Digital Startups: the mediating role of entrepreneurship. International Journal of Human Capital in Urban Management, 8(1), 17-30. doi: 10.22034/IJHCUM.2023.01.02
Investigating social capital, trust and commitment in family business: case of media firms. Journal of Family Business Management. https://doi.org/10.1108/JFBM-02-2021-0013
- The two articles were added to the literature review section. I appreciate the reviewer for the recommendation.
Best of luck with the further development of the paper.
- Again, we highly appreciate your valuable comments.
Comments on the Quality of English Language
-Another round of spellchecking by a native speaker is recommended.
- After revision, we let our paper be edited by a professional editing service.

Reviewer 3 Report
Manuscript Number: behavsci-2363076
Title: Impact of Superiors’ Ethical Leadership on Subordinates’ Unethical Pro-Organizational Behavior: Mediating Effects of Followership
Due Date: 30 April, 2023
COMMENTS TO THE AUTHORS
________________________________________________________________
Dear Author/s,
Thank you for submitting your manuscript to Behavioral Sciences. It is an interesting read but there are a number of issues that require your attention before this paper can be published. Some are minor and can easily be addressed, others require some further reflection and effort, as indicated below:
Your paper needs some improvement. More specific areas of focus are as follows:
Introduction, Literature Review and Research Question/s
The introduction sets the focus for the paper adequately. Perhaps defining ethical behavior could be useful since it is referred to throughout the document.
Although attempts are made to consider different aspects of the subject matter, the paper does not come across strongly from the point of view of critical discussion of the different theoretical constructs. Perhaps a discussion juxtaposing ethical aspects in relation to leadership and behavior should come further up in the discussion so that the unethical behavior referred to in the paper can be compared to the standard, moral behavior typically expected in organisations.
Good conduct and moral judgement need to be further unpacked to set the benchmark of what ethical behavior should entail.
Motivation and Contributions
This area is weak. The literature review could benefit from further engagement with the literature. It would be useful to further establish, perhaps more explicitly what the contributions of the paper to the literature are.
Theoretical Foundations, Model and Hypotheses
The conceptual model is clearly articulated and the hypotheses are clearly stated. It could help if further link to the literature is made although one can appreciate that references may not be easy to find in this area.
Methods
This section is comprehensive. Perhaps there could be more clarity as to how the ages where categorized.
Ethical considerations appear to be missing. It is important to add reference to the ethical considerations made and the permissions acquired to access the research companies.
Measures: It could be useful for the reader to add how many items there are in each of the scales used and if any adaptations have been made. The final version of the survey could be added to the supplementary materials.
Results.
The results are clearly presented.
Discussion – this section is missing.
Although the section per se is missing, the conclusion is extensive. The authors could consider splitting the discussion, limitations to the study and conclusion for better structuring of the paper.
General comments.
In general, the topic is interesting and the paper is relatively well positioned. Adding the suggested changes could make the paper more robust.
Best wishes for further developing your paper.
adequate
Author Response
Response to the Reviewer 3
We would like to express our appreciation to the peer reviewer who provided thoughtful and scholarly evaluations of our manuscript. We tried to apply all your suggestions. We marked the revised part of the manuscript in green.
Introduction, Literature Review and Research Question/s
The introduction sets the focus for the paper adequately. Perhaps defining ethical behavior could be useful since it is referred to throughout the document.
Although attempts are made to consider different aspects of the subject matter, the paper does not come across strongly from the point of view of critical discussion of the different theoretical constructs. Perhaps a discussion juxtaposing ethical aspects in relation to leadership and behavior should come further up in the discussion so that the unethical behavior referred to in the paper can be compared to the standard, moral behavior typically expected in organisations. Good conduct and moral judgement need to be further unpacked to set the benchmark of what ethical behavior should entail.
- We added the definition of ethical behavior as a footnote in the introduction.
“Ethical behavior in an organization refers to actions and decisions made by individuals within the organization aligned with moral and societal standards [17]. It involves making choices that consider the well-being of all stakeholders, including employees, customers, shareholders, and the public [18]. Ethical behavior also requires complying with legal and regulatory requirements and responsibility in all aspects of work.”
Motivation and Contributions
This area is weak. The literature review could benefit from further engagement with the literature. It would be useful to further establish, perhaps more explicitly what the contributions of the paper to the literature are.
- We have summarized the theoretical contributions of this paper as follows.
“The findings of this study contribute to the literature on leadership and organizational ethics in multiple ways. First, it contributes to existing research on ethical leadership by introducing the concept of followership and examining its role in promoting ethical consciousness. Identifying the mutual influence between leaders and followers, especially the role of followers in promoting ethical conduct within an organization, is a crucial gap in this field [20–21].
Second, this study was among the first to investigate the mediating role of followership in leadership and UPB. Contrary to the traditional leader-centric approach in the leadership literature, this study takes a role-based followership approach and examines the development of organizational ethics by considering subordinates and their characteristics.
Third, this empirical study was conducted in a public organizational setting. Existing research on ethical consciousness has mostly focused on the corporate sector, and detailed empirical analysis of ethical consciousness in the public sector is needed [23 – 25]. It is essential to acknowledge the differences in characteristics between public and private organizations and to examine them separately. Through a critical analysis of ethical leadership and unethical behavior, this study aims to expand the scope of research beyond corporate-oriented research and provide valuable insights for promoting ethical behavior in the public sector.”
Theoretical Foundations, Model and Hypotheses
The conceptual model is clearly articulated and the hypotheses are clearly stated. It could help if further link to the literature is made although one can appreciate that references may not be easy to find in this area.
- We added the following 16 relevant references that may strengthen our conceptual model and hypotheses and applied the references throughout the paper.
- Lord, R.G., Epitropaki, O., Foti, R.J., Hansbrough, T.K. Implicit leadership theories, implicit followership theories, and dynamic processing of leadership information. Annual Review of Organisation Psychology and Organizational Behavior. 2020. 1, 49-74. https://doi.org/10.1146/annurev-orgpsych-012119-045434
- DeRue, D.S., Ashford, S.J. Who will lead and who will follow? A social process of leader identity construction in organizations. Academy of Management Review. 2010. 35:627–47. https://doi.org/10.5465/AMR.2010.53503267
- Steffens, N.K., Munt, K.A., van Knippenberg, D., Platow, M.J., Haslam, S.A. Advancing the social identity theory of leadership: a meta-analytic review of leader group prototypicality. Organizational Psychology Review. 2021. 11, 35-72, doi: 10.1177%2F2041386620962569.
- Carsten, M.K., Uhl-Bien, M. Ethical followership: an examination of followership beliefs and crimes of obedience. Journal of Leadership and Organizational Studies. 2013.20. 49-61, doi: 10.1177/1548051812465890.
- Lu, CS., Lin, CC. The Effects of Ethical Leadership and Ethical Climate on Employee Ethical Behavior in the International Port Context. J Bus Ethics. 2014. 124, 209–223. https://doi.org/10.1007/s10551-013-1868-y
- Pio, R.J., Lengkong, F.D.J. The relationship between spiritual leadership to quality of work life and ethical behavior and its implication to increasing the organizational citizenship behavior. Journal of Management Development. 2020. 39, 293-305. https://doi.org/10.1108/JMD-07-2018-0186
- Tajpour, M., Razavi, S. The effect of team performance on the internationalization of Digital Startups: the mediating role of entrepreneurship. International Journal of Human Capital in Urban Management.2023. 8, 17-30. doi: 10.22034/IJHCUM.2023.01.02
- Tajpour, M., Salamzadeh, A., Salamzadeh, Y., Braga, V. Investigating social capital, trust and commitment in family business: case of media firms. Journal of Family Business Management. 2020. 12, 938-958. https://doi.org/10.1108/JFBM-02-2021-0013
- Alniacik, E., Kelebek, E. F. E., Alniacik, U. The moderating role of message framing on the links between organizational identification and unethical pro-organizational behavior. Management Research Review. 2021. 45, 502-523. https://doi.org/10.1108/MRR-01-2021-0004
- Baur, C., Soucek, R., Kuhnen, U., Baumeister, P. F. Unable to resist the temptation to tell the truth or to lie for the organization? Identification makes the difference. Journal of Business Ethics. 2020. 167, 643–662. https://doi.org/10.1007/s10551-019-04162-3
- Irshad, M., Bashir, S. The dark side of organizational identification: A multi-study investigation of negative outcomes. Frontiers in Psychology. 2020. 11, 2521. https://doi.org/10.3389/fpsyg.2020.572478
- Kim, M. Y., Miao, Q., Park, S. M. Exploring the relationship between ethical climate and behavioral outcomes in the Chinese public sector: The mediating roles of affective and cognitive responses. International Journal of Business, Humanities and Technology. 2015. 5, 88–103.
- Miao, Q., Newman, A., Yu, J., Xu, L. The relationship between ethical leadership and unethical pro-organizational behavior: Linear or curvilinear effects?. Journal of Business Ethics. 2013. 116, 641–653. https://doi.org/10.1007/s10551-012-1504-2
- Naseer, S., Bouckenooghe, D., Syed, F., Khan, A. K., Qazi, S. The malevolent side of organizational identification: Unraveling the impact of psychological entitlement and manipulative personality on unethical work behaviors. Journal of Business and Psychology. 2020. 35, 333–346. https://doi.org/10.1007/s10869-019-09623-0
- Wang, Y. J., Li, H. Moral leadership and unethical proorganizational behavior: A moderated mediation model. Frontiers in Psychology. 2019. 10, 1–19. https://doi.org/10.3389/fpsyg.2019.02640
- Zhang, X. C., Yao, Z. Impact of relational leadership on employees' unethical pro-organizational behavior: A survey based ontourism companies in four countries. PLoS ONE. 2019. 14(12), 1–19. https://doi.org/10.1371/journal.pone.0225706
Methods
This section is comprehensive. Perhaps there could be more clarity as to how the ages where categorized.
- Subordinates’ and superiors’ ages were measured in the following order: below 30 (1), 30-39 (2), 40-49 (3), and over 50 (4).
Ethical considerations appear to be missing. It is important to add reference to the ethical considerations made and the permissions acquired to access the research companies.
- We added the following in the data collection section.
“Ethical clearance for the study was sought from the committee of the university where the first author was employed. All ethical considerations for the research involving human subjects, including informed consent and anonymity, were upheld. Participation in the survey was voluntary. Survey participants could leave any questions they did not wish to answer unanswered. They were also allowed to refuse to participate or withdraw at any time. Their answers remained anonymous and confidential.”
Measures: It could be useful for the reader to add how many items there are in each of the scales used and if any adaptations have been made. The final version of the survey could be added to the supplementary materials.
- We adapted the questionnaire to measure ethical leadership, followership, and UPB to fit the public institution context. We added explanations in 3.1 Questionnaire Formulation section. And Scales were added as supplementary material.
Results.
The results are clearly presented.
Discussion – this section is missing.
Although the section per se is missing, the conclusion is extensive. The authors could consider splitting the discussion, limitations to the study and conclusion for better structuring of the paper.
- We split the conclusion into theoretical and practical implications, limitations, future suggestions, and conclusion sections. Also, the contents were sufficiently added in each section.
General comments.
In general, the topic is interesting and the paper is relatively well positioned. Adding the suggested changes could make the paper more robust.
Best wishes for further developing your paper.
Again, we highly appreciate your valuable comments.
Comments on the Quality of English Language
Adequate
-After revision, we let our paper be edited by a professional editing service.

Round 2
Reviewer 2 Report
Dear author(s)
Hope you are doing well. According to the review of this article, the corrections have been made.
Good luck
Author Response
Again, we thank the reviewer for our manuscript's thoughtful and scholarly evaluations.